# Mechanisms of Neurodegeneration in Various Forms of Parkinsonism—Similarities and Differences

**DOI:** 10.3390/cells10030656

**Published:** 2021-03-16

**Authors:** Dariusz Koziorowski, Monika Figura, Łukasz M. Milanowski, Stanisław Szlufik, Piotr Alster, Natalia Madetko, Andrzej Friedman

**Affiliations:** 1Department of Neurology, Faculty of Heath Science, Medical University of Warsaw, 03-285 Warsaw, Poland; monika.figura@wum.edu.pl (M.F.); lukasz.milanowski@wum.edu.pl (Ł.M.M.); stanislaw.szlufik@wum.edu.pl (S.S.); piotr.alster@wum.edu.pl (P.A.); natalia.madetko@wum.edu.pl (N.M.); andrzej.friedman@wum.edu.pl (A.F.); 2Department of Neurology, Mayo Clinic, Jacksonville, FL 32224, USA; 3Department of Neuroscience, Mayo Clinic Florida, Jacksonville, FL 32224, USA

**Keywords:** neurodegeneration, PD, PSP, MSA, DLB, *SNCA*, *MAPT*, oxidative stress, mitophagy

## Abstract

Parkinson’s disease (PD), dementia with Lewy body (DLB), progressive supranuclear palsy (PSP), corticobasal degeneration (CBD) and multiple system atrophy (MSA) belong to a group of neurodegenerative diseases called parkinsonian syndromes. They share several clinical, neuropathological and genetic features. Neurodegenerative diseases are characterized by the progressive dysfunction of specific populations of neurons, determining clinical presentation. Neuronal loss is associated with extra- and intracellular accumulation of misfolded proteins. The parkinsonian diseases affect distinct areas of the brain. PD and MSA belong to a group of synucleinopathies that are characterized by the presence of fibrillary aggregates of α-synuclein protein in the cytoplasm of selected populations of neurons and glial cells. PSP is a tauopathy associated with the pathological aggregation of the microtubule associated tau protein. Although PD is common in the world’s aging population and has been extensively studied, the exact mechanisms of the neurodegeneration are still not fully understood. Growing evidence indicates that parkinsonian disorders to some extent share a genetic background, with two key components identified so far: the microtubule associated tau protein gene (*MAPT*) and the α-synuclein gene (*SNCA*). The main pathways of parkinsonian neurodegeneration described in the literature are the protein and mitochondrial pathways. The factors that lead to neurodegeneration are primarily environmental toxins, inflammatory factors, oxidative stress and traumatic brain injury.

## 1. Introduction

Neurodegenerative diseases are currently one of the core medical problems due to the general demographic composition of the population. A better understanding of the mechanisms leading to the development of neurodegeneration at the cellular and molecular level can lead to effective prevention and treatment in the future. The pathogenesis of movement disorders associated with parkinsonism and dementia have not yet been fully established. Parkinson’s disease (PD) is the second most common neurodegenerative disease after Alzheimer’s disease (AD). The group of diseases belonging to atypical parkinsonism is less common and includes disorders such as dementia with Lewy body (DLB), progressive supranuclear palsy (PSP), corticobasal degeneration (CBD) and multisystem atrophy (MSA) [1,2,3,4,5].

All parkinsonisms are characterized by a similar clinical picture, especially at the beginning of the disease. Certain units, such as PD and parkinsonian variants of MSA and PSP, are indistinguishable in the first phase of the disease. However, they are characterized by different neuropathologies. The common problems are the lack of clearly defined mechanisms leading to neurodegeneration and the overlapping of neuropathological processes in the final stage of the disease [6]. This may indirectly prove that all neurodegenerative processes have some similarities. Various forms of parkinsonism share several clinical, neuropathological and genetic features. Parkinsonism is a clinical symptom characterized by combinations of bradykinesia, rigidity, postural instability, flexed posture, freezing phenomenon and tremor. Specific combinations of non-motor symptoms, such as autonomic dysfunction, cerebellar signs, neuropsychiatric symptoms and ocular movement abnormalities, characterize each particular parkinsonian disorder. The parkinsonian syndromes are grouped together based on their shared clinical features, but are separated on the basis of their different pathologies. Parkinsonism is characterized by selective neuronal loss in different brain regions and the presence of abnormal inclusion bodies in the cells of the brain tissue. Loss of dopaminergic neurons is a common feature shared by all parkinsonian disorders.

Based on the results of immunochemical staining of the inclusion bodies, the diseases can be divided into two groups: synucleinopathies and tauopathies. PD, MSA and DLB belong to the synucleinopathies, which are characterized by the presence of fibrillary aggregates of α-synuclein protein in the cytoplasm of selected populations of neurons and glial cells. PSP is a tauopathy associated with the pathological aggregation of the microtubule-associated tau protein. Growing evidence indicates that parkinsonian disorders to some extent share a genetic background, with two key factors identified so far: the microtubule-associated tau protein gene (*MAPT*) and the α-synuclein gene (*SNCA*). Below, we present a brief description of individual disease entities. The neuropathological data are presented in Table 1.

PD affects more than 1% of those over the age of 60 and up to 4% of those over the age of 80. The average age of onset is around age 60. There is no cure for PD. Clinical criteria require the presence of bradykinesia and any one of typical rest tremor, extrapyramidal rigidity or postural instability [7]. The disease motor symptoms mainly result from a selective degeneration of dopaminergic neurons within the substantia nigra (SN) and the consequent reduction of the neurotransmitter dopamine. In addition to motor symptoms, non-motor symptoms may occur, such as hyposmia, autonomic dysfunction, sleep disorders, cognitive impairment and depression

DLB is the second most common type of progressive dementia after Alzheimer’s disease. DLB and PD dementia (PDD) are clinically similar and share characteristic neuropathological changes. The timing of dementia relative to parkinsonism is the major clinical distinction between DLB and PDD, with dementia arising in the setting of well-established idiopathic PD (after at least 1 year of motor symptoms), denoting PDD, while earlier cognitive impairment relative to parkinsonism denotes DLB [8]. Current clinical diagnostic criteria emphasize these features and also weigh the evidence for dopamine cell loss measured with single-photon emission computed tomography (SPECT) imaging and for rapid eye movement (REM) sleep behaviour disorder, a risk factor for synucleinopathies [2]. DLB and PDD are important and common dementia syndromes that overlap in their clinical features, neuropathology and management, with some controversy in their differentiation [9].

PSP is a rare neurodegenerative disorder (prevalence is 3.1–6.5/100,000) characterized by the presence of tau-immunoreactive inclusion bodies in the neurons and glial cells. Unexplained falls are an early sign of the disease. Eventually patients present with postural instability, vertical gaze palsy, axial rigidity and cognitive impairment

CBD is a rare progressive neurodegenerative disorder with a prevalence of approximately 6 cases per 100,000 people [10]. Age of disease onset is most commonly between 40 and 70, and the average survival time is 7 years [11]. The definite diagnosis of CBD can only be performed neuropathologically as its clinical manifestation may overlap with other neurodegenerative disorders (PSP, AD and frontotemporal dementia), which makes a clinical diagnosis very challenging [11,12,13].

MSA is an adult-onset clinically progressive neurodegenerative disease of unknown aetiology [14]. It is characterized by a variable combination of parkinsonism, cerebellar ataxia and autonomic dysfunction. MSA has a prevalence of about 2–5 per 100,000 people. MSA is a synucleinopathy that affects not only the nigrostriatal dopaminergic pathway, but also the cerebellar afferent pathways (pontocerebellar and olivocerebellar fibres). On brain autopsy in MSA with predominant parkinsonism, brownish discoloration of the posterolateral putamen can be observed. Significant cerebellar signs include atrophy of the pontine base, the cerebellar white matter and, to a lesser extent, the medulla (e.g., inferior olive) and the cerebellar cortex. The histopathologic hallmark of MSA is accumulation of α-synuclein within glial cytoplasmic inclusions (GCIs). The main pathologic changes include striatonigral degeneration and olivopontocerebellar atrophy [15].

The mechanism of neurodegeneration is usually a combination of genetic and environmental factors. The most important processes include protein pathology, mitochondrial pathology via genetic factors, inflammation and oxidative stress. These pathways are often disturbed by toxic factors and traumatic brain injury (Figure 1). Neurodegeneration with various pathological pattern may affect different areas of the brain results in specific for the localization clinical phenotypes. The selective degradation of some type of neurons (nigrostriatal/striatal) cause that the parkinsonism is observed in all individuals. 

**Table 1 cells-10-00656-t001:** Main neuropathological features of the different parkinsonisms.

Location	Type of Pathology	Macroscopic Localisation	Microscopic Changes	Localisation	Spreading
**Parkinson Disease**					
CNS	LB and LN	SN, LC	Neuronal loss of neurons	Dopaminergic neurons of SN; neuronal loss of noradrenergic neurons in the LC, neuronal cell bodies—synapses, axons and astroglial cells [16,17,18]	Early: dorsal motor nucleus of the vagus and the anterior olfactory nucleus Medium: LC and SN Later: basal forebrain, amygdala, medial temporal lobe structures, and cortical areas
PNS	α-synuclein aggregates [19], phosphorylated α-synuclein inclusions [20]	Enteric, pelvic and cardiac ganglia [21,22,23,24,25,26,27] skin [28], pharyngeal motor and sensory branch of the vagus nerve, glossopharyngeal nerve, internal superior laryngeal nerve [29,30]	Axonal degeneration after traumatic nerve injury [31] small fibre neuropathy [32]	The spinous cell layer, pilosebaceous unit and eccrine glands [28], sympathetic ganglia and intermediolateral column of the medulla [27], distal cardiac sympathetic axons [25] gastrointestinal tract: 90% as neurites, 10% soma [33,34,35]	Peripheral-to-central spreading pattern of in sympathetic nervous system [36], rostro-caudal gradient in gastro-intestinal tract [37], spreading from vagal terminals of the gut to dorsal motor nucleus of vagus nerve [38,39]
**Dementia with Lewy Bodies**					
CNS	LB, LN, α-synuclein aggregates in oligodendrocytes [40,41]	Diffuse neocortical, limbic, brainstem, amygdala, olfactory bulb, SN [42]	Neuronal loss of neurons	Diffuse neocortical, limbic, brainstem, amygdala, olfactory bulb, SN [42]	Due to Braak stages
**Progressive Supranuclear Palsy**					
CNS	Accumulation of tau inclusions in neurons	Marked atrophy of the midbrain and superior cerebellar peduncle along with mild frontal cortical atrophy; STN smaller than normal	Inclusion bodies in astrocytes and in oligodendroglia	GP, STN, midbrain tectum, periaqueductal gray, LC, cerebellar dentate nucleus, corpus striatum, ventrolateral thalamus, red nucleus, pontine and medullary tegmentum, pontine base, inferior olivary nucleus	Due to Braak stages
**Corticobasal Degeneration**					
CNS	Deposition of tau in neurons and glia [10,11,12,13,43]	Disproportionately more in forebrain structures, than in hindbrain	Pretangles, NFT, neuropil threads, astrocytic plaques, oligodendroglial coiled bodies [10,43]	Forebrain structures > hindbrain	Due to Braak stages
**Multiple System Atrophy**					
CNS	α-synuclein inclusions [44]	Striatonigral degeneration and olivoponto-cerebellar atrophy	Accumulation of α-synuclein within GCIs, neuronal cytoplasmic inclusions [45,46]	Oligodendroglial cells, neurons [47,48]	
PNS	Filamentous α-synuclein aggregates [49]	Multidomain autonomic nervous system failure [50,51,52], reduction of sensory afferent and postganglionic sympathetic fibres [53]	Cholinergic, catecholaminergic, noradrenergic, serotonergic preganglionic, postganglionic neurons [54,55,56,57], postganglionic fibres [58]	Cholinergic neurons in dorsal motor nucleus and ventrolateral nucleus ambiguous of the vagus [55], catecholaminergic neurons of ventrolateral medulla [54], medullary arcuate nucleus [59], noradrenergic LC [56], medullary serotonergic groups, ventrolateral medulla [60,61], ventromedullary NK-1-receptor-immunoreactive neurons [62], A5 noradrenergic neurons [57], caudal raphe nucleus with sparing of rostral raphe neurons [63,64], Edinger-Westphal nucleus and posterior hypothalamus [65], suprachiasmatic nucleus [66], pontomedullary reticular formation [46,67] sympathetic preganglionic neurons in intermediolateral column of thoracolumbar spinal cord [56,68], postganglionic sudomotor nerves [69], cardiac postganglionic sympathetic fibres [58], Schwann cells’ cytoplasm [49]	

CNS—central nervous system; PNS—peripheral nervous system; LB—Lewy bodies; LN—Lewy neurites; SN—substantia nigra; LC—locus ceruleus; GCIs—glial cytoplasmic inclusions; STN—subthalamic nucleus; NfT—neurofibrillary tangles; GP—globus pallidus.

## 2. Genetics of Parkinsonism

The most common parkinsonism in which genetic factors play an important role is PD—a mostly sporadic disease, but about 10–15% of cases are monogenic forms [70]. The first gene identified as associated with PD was *SNCA* point mutations, then multiplications were identified [71,72]. This discovery also led to linking PD pathophysiology with α-synuclein. There are 23 genes associated with monogenic PD (Table 2). There are three main ways in which they are inherited: autosomal dominant (*LRRK2*, *SNCA*), autosomal recessive (*PRKN*, *PINK1* and *DJ1*) or X-Linked recessive (*TAF1*). About 78 genetic loci are connected with PD risk genes, which can potentially increase the chances of PD occurrence over a lifetime [73]. They represent different pathophysiology and include pathways associated with mitochondria quality control (*PRKN*, *PINK1*, *FBXO7*, *EIF4G1*, *HTRA2*, *CHCD2* and *DJ1*), synuclein pathophysiology (*SNCA* and *LRRK2*), lysosomal enzymes (*GBA* and *ATP13A2*), oxidative stress (*DJ1*) and ubiquitin activity (*PRKN* and *PINK1*).

There are some characteristic clinical symptoms in monogenic forms. For example, in autosomal recessive genes associated with the mitochondrial quality control pathway, early age of onset (<50 years old), tremor dominant PD and good response for levodopa are typical. There is some evidence that the impact of the disease-causing mutations is earlier and occurs even in the premotor phase.

Not only genomic DNA is important in the pathophysiology of PD. The regulation of gene expression by microRNA (miRNA) seems to play a significant role in many neurological disorders. Several of them were associated with PD [74,75].

Genetic studies of other parkinsonisms are not as common as for classical PD. Possible reasons may be the rarity of these diseases and the difficulty in diagnosis. For other synucleinopathies there have been only single GWAS studies with a low number of participants [76]. In DLB many studies were conducted using clinically diagnosed patients only. The most frequently analysed candidate genes were those previously associated with other neurodegenerations, such as *TREM2*, *C9orf72*, *APOE*, *GBA*, *MAPT* and all PD genes responsible for monogenic forms of PD (Table 2) [76]. In a cohort of 102 clinically diagnosed DLB patients, 2 patients had more than 30 *C9orf72* repeats, which was not observed in other large-cohort studies [77,78,79]. The *TREM2* (the triggering receptor expressed on myeloid cells 2, a well-established AD risk factor) *R62H* and *GRN* variants, which are rare, were suggested as being associated with DLB, but another study did not find any similar association with the *TREM2 R47H* variant [80,81]. The only GWAS study performed on 1324 pathologically confirmed DLB patients revealed associations with *SNCA*, *APOE* and *GBA* [82].

Only a few genetic studies were conducted in MSA patients. A positive association was reported with the *COQ2*, *SNCA* and *MAPT* genes [83,84,85]. However, a single GWAS study of 918 MSA Caucasian patients found an association with the *ELOVL7*, *FBXO47*, *EDN1* and *MAPT* loci, but not with *COQ2* or *SNCA* [86]. Genes previously associated with spinocerebellar ataxias were also analysed. In a study of 80 clinically diagnosed MSA patients, 22 and 12 CAG repeats in the SCA6 gene were found in one 58-year-old female [87]. The last study by Wernick et al. revealed an increased risk of MSA associated with more than 38 CAG/CAA repeats in the *TBP* gene causing SCA17. This gene encoded the general transcription factor, TATA-binding protein, in which mutations may be associated with improper mRNA synthesis [88].

Tauopathies are even less represented in genetic studies compared to synucleinopathies. The best characterized association with PSP occurrence is mutations in *MAPT*. Alternative splicing of exon 10 produces two major isoforms of the tau protein, namely, 4R-tau and 3R-tau, with four and three microtubule-binding repeats. PSP is the most often reported pure 4R tauopathy. Several point mutations in exon 10 have been discovered. The most important is the *N279K* mutation, which was reported in a series of a few cases of familial PSP [89]. Another important observation in PSP is that haplotype H1, which has 11 TG repeats with 238 bp in intron 9, is overrepresented in PSP patients compared to controls [90]. Similar observations were also confirmed in a GWAS study [91]. A positive correlation with *MAPT* was also observed in another tauopathy—CBD [92]. The rarest cause of parkinsonism are mutations in exon 2 of the *DCTN1* gene. There were several point mutations in the Cap-Gly domain described in 26 families worldwide. *DCTN*1 is important in proper microtubule transport and mutations of it are responsible for Perry syndrome, with a unique TDP43 pathology [93].

## 3. Protein Pathology in Various Forms of Parkinsonism

### 3.1. Synucleinopathies

#### Propagation of PD-Related Synucleinopathy

α-synuclein is natively a mostly unfolded protein that participates in transmembrane transport. It can be found in the presynaptic terminals of neurons. Defects in synaptic vesicle exocytosis have been observed in α-synuclein transgenic mice and experiments using α-synuclein knockout mice indicate that α-synuclein deletion leads to increased dopamine release [94,95]. 

At the beginning of the neurodegenerative process, α-synuclein changes its conformation from a soluble α-helical structure to a SN-sheet-rich structure. This further polymerizes to form toxic oligomers and amyloid plaques. Oligomeric soluble forms of α-synuclein, invisible on histopathological examination, seem to play a major role in proteasome and lysosomal mechanisms [96]. They propagate with axonal transport and lead to proteostasis imbalance in nerve cells [97,98]. These oligomers then form fibrils and then α-synuclein deposits—LB. The Ser-129 phosphorylated form of α-synuclein seems to be particularly important in LB formation, where it constitutes over 90% of total α-synuclein vs. only 4% in a healthy brain [99]. 

The intraneuronal propagation of α-synuclein deposits is not random. In most cases it follows a specific pattern described by Braak et al. [100]. The initiation of the disease is according to the “dual hit” hypothesis postulated by the authors—triggered in the olfactory bulb and the autonomic gastroenteric plexuses, and then ascending to the central nervous system [101]. Numerous histopathological studies confirmed multi-organ distribution of LB and LN in the gastroenteric plexus, salivary glands, adrenal glands and cardiac plexuses [27].

The mode of α-synuclein transport is different for each form. While monomeric α-synuclein is released from nerve terminals in the striatum via ATP-dependent K+ channels, the oligomeric form and fibrils are thought to be released via exocytosis. This process may be increased by lysosomal dysfunction. Around 5% of extracellular oligomeric α-synuclein is associated with exosomes, both on the inside and surface of the vesicles [102,103,104,105]. 

Due to its mechanism of propagation, α-synuclein was suggested by some authors to act as a prion [106]. It shares some features with prion diseases, such as propagation by misfolding into an SN-sheet conformation with the formation of toxic oligomers and amyloid aggregates. These then serve as a template to promote conformational change in the wild-type protein and the neurodegenerative process spreads. In relation to PD, the prion theory suggested that foetal dopaminergic neuronal grafts develop an LB pathology in post-mortem brain studies of PD patients treated with a transplant [107]. Lee et al. indicated that injection of brain extracts prepared from DLB patients induced the deposition of α-synuclein aggregates in myenteric neurons [108]. Consistently, some evidence for the neuroprotective effect of total vagotomy against PD was described based on an observation of a Swedish cohort of patients [109].

There is not, however, enough evidence to support the prion theory in PD. Firstly, there is no evidence for transmission of PD from one individual to another. Synucleinopathy propagation happens in a selective way, sparing specific types of neurons, even within one nucleus [110]. Post-mortem analysis of PD brains has revealed that the majority of LB and LN occur within the pigmented neurons in the SNc. Synaptic expression of α-synuclein is mostly accompanied by expression of vesicular glutamate transporter-1, an excitatory synapse marker protein. In contrast, α-synuclein expression in inhibitory synapses differs between brain regions. As endogenous physiological expression of α-synuclein is required for aggregate formation, neurons with a low natural expression of α-synuclein (inhibitory neurons expressing glutamic acid decarboxylase, parvalbumin or somatostatin) exhibit a lower expression of LB [111]. Apart from the staging of the disease based on post-mortem examinations of the patients, all information on α-synuclein propagation has come from experimental animals and not from humans. Some authors suggest, therefore, naming α-synuclein “prion-like” or “prionoid” [112].

A pivotal role in the α-synuclein aggregation process belongs to the central segment of α-synuclein, residues 68–78 of the non-amyloid-β component (NAC). NAC includes an 11-residue segment—NACore. Rodriguez et al. suggest this segment plays a critical role in both the aggregation and the cytotoxicity of α-synuclein. They also conclude that NACore fibrils are comparably toxic to similarly aggregated full α-synuclein [113]. 

### 3.2. Mechanisms of α-Synuclein Toxicity in PD

#### 3.2.1. Microglial Activation

Pathological α-synuclein triggers neurodegeneration in many ways [114]. Firstly, it contributes to microglia activation. Substantial evidence suggests that there is a relationship between α-synuclein oligomerization and the generation of ROS, release of proinflammatory cytokines and nitric oxide among others [115,116,117,118]. Activated microglial cells absorb α-synuclein to remove it from the extracellular space by apoptotic neuron death, autophagy or mechanisms of cellular release. Guo et al. provide evidence from in vitro models that microglia may not only be activated by the presence of α-synuclein, but also that it may in return facilitate its exosomal transmission to other neurons [119].

#### 3.2.2. Synaptic Dysfunction

Synuclein plays a role in the physiological release of neurotransmitters from presynaptic vessels by promoting SNARE-complex assembly [120]. Synaptic synuclein aggregation is supposed to be the initial, important step in synucleinopathy development. α-synuclein overexpression can alter the size of the synaptic vesicle pools, impair their trafficking, misregulate or redistribute proteins of the presynaptic SNARE complex and reduce the endocytic retrieval of synaptic vesicle membranes during vesicle recycling [121,122]. DA neurons may be more vulnerable to synuclein accumulation, based on observations from animal models [123].

#### 3.2.3. Mitochondrial Dysfunction

There is growing evidence to suggest α-synuclein can affect mitochondrial dynamics. Some authors suggest that it can interact with the outer mitochondrial membrane or that it may interact with the F-type ATPase [124,125]. Therefore, it is possible that mitochondrial dysfunction and α-synuclein deposits are not only co-existing but are also co-dependent phenomena. α-synuclein can damage mitochondrial fusion mechanisms and lead to mitochondrial fragmentation [126]. Di Maio et al. also propose that some α-synuclein types (e.g., oligomeric, DA-modified and Ser-129 phosphorylated) can inhibit mitochondrial protein imports in vitro. The authors also mention that this may lead to a decreased respiratory capacity of mitochondria [125].

#### 3.2.4. DNA Repair Damage

α-synuclein also plays an important role in the repair of nuclear DNA. Animal studies prove that accumulation of α-synuclein can induce DNA single-strand and double-strand breaks (DSB) [127,128]. Schaser et al. report that Lewy inclusion-containing neurons in both mouse models and human-derived patient tissue demonstrate increased DSB levels. The authors hypothesize that cytoplasmic aggregation of α-synuclein reduces its nuclear levels, increases DSBs, and may lead to programmed cell death via nuclear loss-of-function [129]. Vasquez et al. additionally report that early oligomeric α-synuclein causes more DSB than the monomeric and fibril forms [127]. It is suggested that the oligomeric and aggregated forms of α-synuclein promote genomic instability [130]. The difference in nuclear expression of α-synuclein between the oxidative stress and normal condition still remains unclear [130]. 

#### 3.2.5. Protein Clearance Pathology

α-synuclein aggregation damages both the ubiquitin-proteasome system (UPS) and autophagy-lysosomal pathway (ALP), leading to impaired protein clearance and resulting in its accumulation and aggregation. A decline of both ALP and UPS with age may in turn lead to neurodegeneration [131]. It was observed that lysosomal activity impairment is accompanied by the accumulation of dilated lysosomes in the presence of α-synuclein aggregates [132].

Finally, α-synuclein also acts as a regulator of microtubule formation [133]. It interacts with microtubules and tubulin α2β2 tetramer. Cartelli et al. suggest that PD-linked α-synuclein variants do not undergo tubulin-induced folding and cause tubulin aggregation rather than polymerization [134]. The effect of α-synuclein on the microtubule network could impact the transport and distribution of mitochondria. 

#### 3.2.6. Synucleinopathy in MSA and DLB

Most studies on α-synuclein aggregation and toxicity focus on PD models. It was, however, proved that different synucleinopathies may be caused by different α-synuclein strains. MSA’s hallmarks are GCIs of α-synuclein [135]. GCI-α-synuclein and LB-α-synuclein have different abilities to seed the A53T mutant α-synuclein [136].

Interestingly, in a study by Watts et al., after being injected into a mouse model expressing A53T mutant α-synuclein, GCI-α-synuclein induced an α-synuclein pathology in neurons, but not in oligodendrocytes [137]. This could, however, be caused by the effect of mutation or a naturally very low expression of α-synuclein in oligodendrocytes. The origin of GCI-α-synuclein in glia is unclear. Peng et al. mention two hypotheses of its accumulation in MSA patients—induced expression of α-synuclein in oligodendrocytes or transmission of α-synuclein from neurons to oligodendrocytes [138,139].

The prion-like properties of α-synuclein led to some of the most important findings in understanding differences in the protein between PD and MSA cases. Studies on α-synuclein aggregation in tissues from olfactory mucosa of MSA and PD patients with ultrasensitive Real-Time Quaking Induced Conversion (RT-QuIC) indicated biochemical and morphological features potentially enabling their discrimination [140]. Neurofilament light chain (NfL) is also an important marker in the differentiation between MSA and LB parkinsonism, with higher levels of NfL in MSA [141]. Singer et al. report markedly different reaction kinetics of cerebrospinal fluid (CSF) α-synuclein in MSA and PD/DLB groups in a study using ELISA and protein misfolding cyclic amplification (PMCA) methods. PD and DLB samples resulted in a greater maximum ThT fluorescence than MSA samples, and no difference was recorded between DLB and PD cases [141]. Another PMCA study on α-synuclein strains also confirms that aggregates that are associated with PD and MSA correspond to different conformational strains of α-synuclein [142].

Presynaptic α-synuclein aggregates in the cortex of DLB brain correlate with reduced dendritic spines, suggesting that these aggregates contribute to synapse loss and cognitive dysfunction [143].

A study of CSF-derived exosomes in patients with PD, DLB and other disorders indicate that the level of exosomal α-synuclein is significantly decreased in CSF from patients with DLB compared to neurological controls (among them PD), mainly due to a lower absolute number of CSF exosomes [144].

### 3.3. Tauopathies

Tau is a microtubule-binding protein, mainly expressed in neuronal cells and at lower levels in glia. It localizes predominantly in the axonal tracts of neurons where it stabilizes the microtubules and promotes their assembly [145]. In 1986, tau was first identified as a component of NFT in AD [146].

Mutations in the tau gene led to the determination of the fact that tau dysfunction and accumulation can cause aggregation of hyperphosphorylated tau, microtubule destabilization and impaired protein transport [147,148]. Tau pre-mRNA undergoes alternate splicing in exons 2, 3 and 10, which results in six different isoforms. Of those, 4-repeat isoforms occur in CBD and PSP and 3-repeat isoforms in Pick disease.

Insoluble tau aggregates in PSP and CBD are different from AD. Narasimhan et al. used enriched pathological tau prepared from CBD and AD. The study results show that CBD-tau and AD-tau induce distinct tau pathologies in animal models, with CBD-tau inducing a more oligodendrocyte pathology and AD-tau inducing predominantly a neuronal pathology [149]. Tau in PSP has a shifted ratio of 4R:3R tau. 4R-tau assembles into 13–14 nm straight filaments that aggregate to form NFT in neurons and tufted astrocytes in glial cells [150]. In CBD, aggregates are formed from 4-repeat tau alone. Neuronal NFTs are more dispersed and less argyrophilic than in PSP. Astrocytic plaques are the typical lesions because the aggregated tau is mainly located in cell processes. In contrast, PSP’s tufted astrocytes are laden with tau fibrillary deposits at the soma, with propagation to the cell processes [150]. Despite the identical composition of the tau isoforms in both diseases, post-mortem studies indicate different proteolytic processing of abnormal tau in CBD and PSP cases [151].

Similar to α-synuclein, tau propagation was also compared to prion. Boluda et al. performed injections of PSP brain homogenates on mice. This caused formation of astrocytic aggregates that resemble tufted astrocytes. Consistently, CBD brain lysates induced the formation of pathology reminiscent of astrocytic plaques [152].

Two basic concepts have been proposed to characterize tau pathology progression [145]. These include the prion-like model of propagation, with pathological aggregates of protein inducing a change in the conformation of the proteins in surrounding cells. The other concept involves active cell secretion and transmission of tau-containing exosomes, leading to both extracellular and intraneuronal spread of pathological tau [145].

Tau oligomers have long been acknowledged as pivotal in AD pathology [153]. However, data on tau oligomers in PSP and CBS is scarce. Tau oligomers have recently been identified as having the capacity of seeding the oligomerization of both the 3R and 4R tau isoforms. There are higher levels of tau oligomers in PSP brain compared to healthy controls [154]. Factors such as the presence of free fatty acids, heparin, RNA or hyperphosphorylation may facilitate aggregation of tau released from microtubules [155]. α-synuclein has been shown to affect the phosphorylation state of tau and induce its fibrillization [156,157,158].

In AD, elevated CSF total tau and phosphorylated tau 181 and decreased β-amyloid1 are useful for diagnosis and may serve as biomarkers. Interestingly, in PSP, despite its strong links to tau, CSF tau and p-tau concentrations are often lower than in age-matched controls, and lower than in AD patients [159]. NfL is another protein considered to be a peripheral biomarker of tauopathies, but not specific. Blood NfL was described as being increased in patients with PSP and CBS, but also MSA, compared to a PD cohort and healthy controls [160]. The CSF NfL/p-tau ratio may be a biomarker of progression of PSP as it is a better predictor of clinical decline compared to phosphorylated tau alone [161].

## 4. Mitochondrial Dysfunction

Mitochondria are double-membrane organelles, which are responsible for respiratory chain and oxidative phosphorylation [1]. The shape, size and number of mitochondria depends on cellular function. Despite the mitochondria having their own DNA (mtDNA), most mitochondrial proteins are encoded by the nuclear genome. In neurons, they also play a role in calcium homeostasis, membrane excitability, neuroplasticity and neurotransmission [162].

Neurons, as cells requiring a lot of energy, are very vulnerable to mitochondrial damage. It can be caused by toxins or genetic factors. 1-methyl-4-phenyl-1,2,3,6-tetrahydropyridine (MPTP), which is a product of contaminants during heroin production, causes parkinsonism and dopaminergic neurons in SN death [163]. It is a by-product of MPTP oxidation, which is a selective inhibitor of mitochondrial respiratory chain complex I [163].

The basic mitochondria function—cellular respiration—implies that they produce many harmful reactive oxygen species (ROS). The proper work of mitochondria is important to protect cells from increased ROS production [164]. To remove damaged organelles, mitochondria conduct a quality-control process called mitophagy, which is a selective degradation by autophagy [165].

The most important proteins in this process are *PINK1* and *Parkin*. *PINK1*, a 581 amino acid protein, is a serine-threonine kinase [166]. *Parkin*, a 465 amino acid protein, is a RING-in-between-RING (RBR)-type E3 ubiquitin (Ub) ligase that catalyses the ubiquitylation of different mitochondrial substrates [167]. Two of the most important molecules associated with both proteins are Mitofusin-1 and Mitofusin-2 [168]. They are mitochondrial fusion proteins that prevent inclusion of damaged mitochondria in the healthy respiratory chain [169]. Mitofusins are removed from mitochondria via the 26S proteasome by a valosin-containing protein (VCP/p97) [169]. It results in fragmentation of damaged mitochondria, which also causes their fusion with autophagosomes [169]. Miro-1 and Miro-2 are two other important proteins that play a significant role in mitochondria trafficking. Damage to these transporter proteins causes inhibition of mitochondrial movement and leads to the accumulation of degenerated mitochondria [170]. Another protein that cooperates with the *PINK1*/*Parkin* pathway in mitophagy is *FBXO7* [171]. It is responsible for *PINK1* downstream in damaged mitochondria by targeting *PRKN* in depolarized mitochondria [171]. DJ-1 is a cysteine protease responsible for protection from oxidative stress and proper mitochondrial function. ROS are reduced by Cys106. It was revealed that in *DJ-1*-associated parkinsonism, patients have an increased percentage of fragmented mitochondria [172].

The present study contributes to understanding the causative mechanisms of MSA. In particular, the observed impairment of respiratory chain activity, mitophagy and coenzyme Q10 biosynthesis suggests that mitochondrial dysfunction plays a crucial role in the pathogenesis of the disease [173]. Furthermore, these findings will hopefully contribute to the identification of novel therapeutic targets for this still incurable disorder.

The impact of tau protein on the proper mitochondrial work was also observed, but mostly analysed only in Alzheimer’s disease models. The mitochondrial function in tau-dependent parkinsonisms require further studies. Phospho tau may impair complex I and result in increased ROS production, loss of Δψm, lipid peroxidation and reduced activities of detoxifying enzymes, such as superoxide dismutase [174]. In P301L, *MAPT* mutants reduced the ATP level and increased susceptibility to oxidative stress was observed [175]. What is more, suppression of complex 1 results in the reduction of ATP levels, which causes redistribution of tau from the axon to the perikaryon [176]. These observations suggest that both mitochondrial damage causes tau accumulation and also that tau accumulation leads to disturbances in proper mitochondrial function.

## 5. Various Neurodegenerative Factors

Proteinopathy and mitophagy are influenced by different environmental and genetic factors that lead to a cascade of processes, resulting in oxidative stress or inflammation. What is more, environmental factors, such as toxins, are strongly related to PD-like damage, and repetitive injuries can lead to parkinsonian disorders, including tauopathy.

### 5.1. Oxidative Stress

For decades, oxidative stress has been considered to be an important factor in neurodegeneration. Mitochondrial dysfunction neurotoxins, such as MPTP, may be responsible for neurodegeneration caused by oxidative stress. The human brain seems to be particularly vulnerable to oxidative stress injury because of its high concentrations of lipids and unsaturated fatty acids, the relatively high concentration of iron and the low concentration of enzymes able to inactivate reactive oxygen species (ROS). The presence of injury due to oxidative stress can be indirectly proven by the presence of high concentrations of oxidation products, e.g., malonic dialdehyde and thymine glycol [177,178].

The mitophagy pathway damage causes not only increased oxidative stress, but also easier iron-dependent a-synuclein oligomerization [179]; it shows that oxidative stress plays a central role in various neurodegenerative processes.

Free radicals are generated, among others, via the Fenton reaction, in which an important role is played by the divalent iron:Fe^+2^ + H_2_O_2_ → Fe^+3^ + OH^−^ + OH**^•^**

Regional differences in the distribution of iron are associated with neurodegenerative diseases involving parkinsonism, and the relationship between iron homeostasis proteins and the regional concentration, distribution and form of iron. Most of the iron in the human brain is present as ferritin-bound iron. It is obvious that any change in the structure or function of ferritin may be related to oxidative damage. The possible role of iron in the process that triggers neurodegeneration via Fenton chemistry is not related to the total iron, but only to the labile iron. Iron measured by electrothermal atomic absorption spectrometry revealed a significant increase in labile iron in PD SN [180]. An important question concerns the source of the iron for the Fenton reaction in PD. In the literature, two possible sources have been considered: neuromelanin and ferritin.

According to several authors, neuromelanin in SN may play a dual role. At the onset of the process it chelates the excess iron, but with the progression of the disease, when the concentration of iron in neuromelanin reaches a critical level, it releases the iron, thus causing oxidative stress. It is difficult to confirm this hypothesis since iron binds to neuromelanin during sample preparation for the determination of iron [181,182]. It may be postulated that in PD SN iron is packed within the ferritin shell in a slightly different way than in the controls, which might be due to a difference in the structure of ferritin between PD and the controls.

There is some controversy in the literature regarding the change in the concentration of ferritin in parkinsonian SN compared to the controls. Faucheux et al. found no difference in the levels of H- and L-ferritins between parkinsonian and control SN pars compacta [183], whereas data by Koziorowski et al. [184], as well as those obtained by Connor et al. [185] on whole SN, showed a considerable decrease in L-ferritin in PD. The pars compacta contains mainly neurons, while pars reticulate has more glial cells. Ferritin in neurons is H-rich, while glial ferritin is L-rich [186]. On the whole, SN samples contain more glial cells and therefore the change in L-ferritin levels might represent changes in these cells [184]. Lower levels of L-ferritin in PD, as in *FTL* gene mutations in neuroferritinopathy, may reduce the ability of ferritin to store iron, resulting in the release of iron. In neuroferritinopathy, the mutations in the *FTL* gene affect protein folding and stability, which increases the intracellular iron availability and sensitivity to oxidative stress and DNA damage [187].

It is generally accepted that oxidative stress plays an important role in the pathogenesis of PD. The consequence of this process is a higher expression of 8-oxoguanine DNA glycosylase (DNA repair enzyme) in SN cases of PD as a cause of DNA oxidative injury [188]. Studies carried out in recent years have permitted the precise characterization of the proteins involved in iron transport. Published data show that downregulation of the iron transporters ferroportin 1 (FP1) and hephaestin (HP) might account for the nigral iron accumulation in 6-hydroxydopamine (6-OHDA)-lesioned animal models and both are involved in cellular iron accumulation and the underlying mechanisms in a cell model of PD [189]. Furthermore, FP1 mutations result in retention of iron in macrophages [190].

An important peptide regulating systemic and brain iron homeostasis is hepcidin. The increase in the synthesis of hepcidin causes iron retention in macrophages and inhibits the absorption of iron in the intestine, thereby lowering the level of plasma iron. Many studies have shown that increased synthesis and secretion of hepcidin is caused by interleukin-6 in the course of an inflammatory response [191]. The effect of other cytokines, acute phase response type 1 (including IL-1α, IL-1β, TNF-α and TNF-β), is indirect and requires more time after the activation of the inflammatory stimulus induces the secretion of IL-6, and it is only IL-6 that induces an increase in the production of hepcidin [192]. Kwiatek-Majkusiak et al. showed that the serum concentrations of hepcidin and IL-6 in the group of all PD patients were significantly higher than in the control group [193]. Additionally, positive correlations between serum hepcidin and IL-6 were found in the PD group [193].

The role for intracellular hepcidin generated by iron in the SN is particularly relevant in restricting iron release by downregulation of FP1 expression in this region and leads to abnormal iron deposition [194]. Published evidence indicate that hepcidin controls iron uptake and release by regulating the expression of the iron transport proteins. Astrocytes treated with hepcidin peptide showed a significant ability to reduce iron uptake and iron release. Moreover, the effect of hepcidin in reducing transferrin receptor 1 expression, which is dependent on the cyclic AMP-protein kinase A pathway, seems to be the primary and dominant event [195].

In the MSA cases, the increased total iron concentration coupled with a disproportionate increase in ferritin in dysmorphic microglia and a reduction in FP expression coexist [196]. This is supported by the isothermal remanent magnetization evidence consistent with elevated concentrations of ferritin-bound iron in MSA basis points [196].

PSP- and CBD-affected neurons exhibit accumulation of NFT derived from the microtubule-associated protein tau, with which ferritin is associated [197]. An increase in the total iron concentration in PSP SN was demonstrated in an early study using inductively coupled plasma spectroscopy, and the possibility that there may be an excess of iron in the basal ganglia of these patients was suggested more recently by a study using electron nanodiffraction and high-resolution electron microscopy [198,199]. Mössbauer spectroscopy showed a higher concentration of iron in PSP in the brain areas involved in this pathological process, SN and GP, compared to a control, while the concentration of iron in pathological tissues in PD did not differ from that in the control [200]. The coefficient of asymmetry of the Mössbauer spectra was significantly higher in GP from PSP than in control GP, which may reflect the different crystallization of iron within ferritin in PSP [200]. In addition, the T1 and T2 relaxation times determined by MRI indicated a dependence of T2 in PSP-GP on factors other than just the concentration of iron [201]. The different behaviour of T2 in PD and PSP may thus be related to different mechanisms of neurodegeneration in these diseases [201].

### 5.2. Neuroinflammation as a Factor in the Neurodegeneration of Parkinsonisms

Nitric oxide is an important factor linking neuroinflammation, oxidative stress and iron metabolism in PD. Microglial cells expressing inducible nitric oxide synthase (iNOS) have been found at an increased density in the SN of PD patients [196]. IL1-β and TNF-α can increase production of NO through expression of CD23 in microglial cells. Superoxide and NO react to form the peroxynitrite anion (ONOO-), which can decompose into an oxidant with reactivity similar to a hydroxyl radical [202]. This phenomenon, with the inhibition of mitochondrial respiratory chain reaction, implicates NO-dependent generation. NO also mediates iron release from ferritin, which leads to iron-related oxidative stress and lipid peroxidation [203].

Some published evidence shows that treatment of mesencephalon neurons by IL-1β or TNF-α leads to increased ferrous iron influx and decreased iron efflux, due to the up-regulation of divalent metal transporter 1 with the iron response element and downregulation of FP1. Increased levels of iron regulatory protein 1, transferrin receptor 1 and hepcidin were also observed in IL-1β- or TNF-α-treated neurons [204].

Several recent studies show that the inflammatory process in PD is not restricted to the central nervous system, but affects also peripheral tissues, including blood. An increased concentration of TNF-α in CSF as well as in the blood serum of PD patients has been reported [205,206,207,208,209]. Two other proinflammatory cytokines: IL-6 and IL-12, have been shown to be at increased concentrations in PD serum [205,206,208,209,210,211,212,213]. Some studies have reported the significance of anti-inflammatory factors, such as IL-10, which are also altered in PD serum [205,213]. We have recently found that a newly characterized proinflammatory factor, NT-proCNP, is increased in PD patients, and that its concentration correlates with TNF-α [205]. Interestingly, *PINK1* upregulates IL-1β-mediated signalling through Tollip and IRAK1 modulation, which links monogenic models of PD with inflammation [214].

Recently, genome-wide association studies (GWAS) provided hypothesis-free evidence of the association of PD with the immunological system. Genes involved in the “regulation of leucocyte/lymphocyte activity” and “cytokine-mediated signalling” were implicated in the disease pathogenesis, contributing to an increased susceptibility to PD.

A genome-wide association study performed in a Caucasian population of PD patients provided evidence of HLA involvement in PD pathogenesis [215]. The association peak was found at single nucleotide polymorphism rs3129882 in the intron of HLA-DRA, which influences the expression of HLA-DR and HLA-DQ. This result was further reinforced by a genome-wide association study performed in a Dutch population and a meta-analysis of genome wide association studies, both pointing to single nucleotide polymorphisms in the HLA II region other than rs3129882 [216,217].

The McGeer et al. work on the mechanism of neurodegeneration provided new insights on the probable association between neuroinflammation and the pathophysiology of parkinsonisms [218]. The neurotoxic activation of microglia is associated with M1-phenotype microglia [219]. The M2-phenotype activation is interpreted as a neuroprotective feature. The role of microglial activation in neurodegeneration was evaluated in various diseases using positron emission tomography (PET) ^11^C-PK11195. The radiotracer was found to be accumulated in the pallidum, midbrain and pons. The abnormalities were found to be correlated with the progress and stage of PSP [220].

A study evaluating examination using combined tau and microglial radiotracers showed colocalization of the pathologic and neuroinflammatory features of the disease. Additionally, the factors ^11^C-PK11195 and ^18^F-AV-1451 showed a positive correlation with the disease’s severity [221]. The pathological basis of PSP, tauopathic inclusions, may be a possible inductor of microglial activation. The microglial activation impacts kinase and phosphatase imbalance, which is related to hyper-phosphorylated tau. The hyper-phosphorylated tau is associated with tau aggregation and tau inclusions [145].

Another hypothesis on the pathophysiology of neurodegeneration in particular is related to astroglial impact [219]. The A2 astrocytes are related to neurotoxic activity in the process of neurodegeneration [219]. On the one hand, it is associated with the induction of synuclein spread [222]. On the other hand, astrocyte dysfunction may be found to be a consequence of elevated iron levels [223]. Recently, interleukin-17A was found to be a factor in the inflammatory pathogenesis of neurodegeneration [224]. A study describing the potential diagnostic role of CCL28, a biomarker of neuroinflammation, was performed in PD and atypical parkinsonisms (AP). In a study by Hall et al., increased levels of this factor were observed in PD. The inflammatory markers serum amyloid A (SAA) and C-reactive protein (CRP) were also found to be significantly increased in MSA and Parkinson’s disease with dementia (PDD) when compared to healthy volunteers and patients affected with PD without dementia [225]. The increase in inflammatory markers was associated with a decline in motor and cognitive abilities. The authors of the study found a correlation between the H&Y stage and changes in CRP, SAA, IL-8 and YKL-40. The observation concerning CRP, SAA and IL-8 was also confirmed in an analysis evaluating correlations with the III part of the Unified Parkinson’s Disease Rating Scale (UPDRS).

In MSA, a study by Harms et al. showed that elevated inflammatory process markers were associated with a more significant deterioration in the clinical course of the disease. The neuroinflammation was associated with α-synucleinopathies in PD, DLB and MSA. A-synuclein was found to play a role in the antigenic epitopes, which activates T-cell response [226]. The neurodegeneration in MSA was associated with T-cell infiltration, which is initiated before neuronal loss [227]. A recent work presents gene enrichment as appearing to be associated with immunity and inflammatory processes in MSA-P and PD. No such observation was confirmed in MSA-C. Deviations were revealed in MSA-P and MSA-C in the context of gene sets of interferon IL-1, IL-6 and IL-8 [228]. Abnormalities in the concentration of cytokines produced by activated microglia as TNF-α, interleukin 1Β and IL-6 were observed in PSP and MSA. IL-2, a regulator of T- and natural killer cells, was found to be deviated in PSP [229]. The MSA immunological abnormal activity is correlated with HLA-DR+ microglial activation in the putamen and SN [230]. The deviation was found to be accompanied by an increase of CD3^+^, CD4^+^ and CD8^+^ in the same regions. Interestingly, an increased salt diet activated inflammation in various diseases, but does not induce neuroinflammation and neurodegeneration in α-synucleinopathies [231]. It is also stressed that the elevated iNOS expression in MSA is correlated with neuroinflammation and neuronal loss [232]. The dysfunction of glial mitochondria is found to be a basis of α-synuclein toxicity in MSA and PD. α-synuclein is interpreted as a factor through which neurons impact mitochondrial dysfunction in the microglia [233]. The analysis of oligodendroglial α-synuclein expression based on experimental models showed overexpression of α-synuclein in the SN [234]. The abnormal expression of the proinflammatory cytokines was also found to be present in PSP. Another work showed that microglial-derived cytokines were more significantly increased in PSP and MSA-P [235]. The changes were not detected in PD [225]. Oxidative stress has been found to be as a possible factor, but the exact mechanism of the impact on the pathogenesis of CBD is not known. A glycoprotein involved in inflammation, progranulin, which is associated with the GRN gene, was interpreted as possibly being correlated with frontotemporal lobar degeneration. [236].

The data presented show that the explored variants of parkinsonism, such as PD, PSP and MSA, share similarities in the pathogenic neuroinflammation, such as an increase in the proinflammatory factors IL-1, IL-6 and TNF-α. The most evident perspective, which is not fully explored, and which may present differences, is the disparity of pathologies that often come up with overlapping neuroinflammatory mechanisms. The similarities and differences in the immunological factors in various forms of parkinsonism are presented in Table 3.

### 5.3. Toxic Neurodegeneration

Neurodegeneration in humans mediated by toxic agents with clinical manifestation of parkinsonian syndrome was first described in the context of MPTP [1,2], which was found in “synthetic heroin”—a narcotic related to meperidine. MPTP itself is harmless, but, as a lipophilic, it can cross the blood-brain barrier. In the central nervous system, MPTP is metabolized to 1-methyl-4-phenylpyridinium (MPP^+^) [237]; this reaction is mediated by monoamine oxidase B (MAO-B) [238]. MPP^+^ is selectively absorbed and concentrated in dopaminergic neurons of the SN pars compacta because of its high affinity to dopamine uptake sites [239]. MPP^+^ is accumulated in mitochondria, where it reaches toxic levels and inhibits complex 1 in the electron transport chain [240]. MPTP also increases the cytosolic dopamine levels, which affects the induction of reactive oxygen species (ROS) and causes cell death [241]. Similar MPTP-induced deficits of mitochondrial complex 1 were described in PD, but not in other neurodegenerative disorders with SN involvement [242]. A study conducted on non-human primates indicates that exposure to MPTP leads to sustained up-regulation of α-synuclein expression in neurons [243].

Rotenone is a highly hydrophobic natural compound extracted from tropical plants and used as a pesticide. It inhibits mitochondrial complex I (at the same site as MPP^+^), generating ROS production and thus causing oxidative stress and cellular death [244].

Paraquat (*N*,*N*-dimethyl-4-4-4-bypiridinium) is a popular herbicide with structural congruity to MPP^+^, a toxic metabolite of MPTP. It penetrates the blood-brain barrier, which is probably mediated by a neutral amino acid transporter [245]. Paraquat selectively destroys midbrain and striatum dopaminergic neurons. The mechanism of paraquat’s neurotoxicity is complex and unique. Paraquat increases glucose uptake with translocation of its transporters to the plasma membrane and seizes control of the pentose phosphate pathway, stimulating its own redox cycling, which generates ROS and impairs the intracellular antioxidant system; this, in turn, decreases glycolysis and impairs the tricarboxylic acid cycle, causing an increase in citrate levels, which has a negative impact on glycolysis through the allosteric inhibition of phosphofructokinase [246]. It has been shown that overexpression of α-synuclein enhances the toxic potential of paraquat by impairing glucose metabolism and mitochondrial metabolic cycles [246]. It has been stated that paraquat neurotoxicity requires microglial activation, which is intensified by iron [247].

Dieldrin, an organochlorine hydrophobic pesticide banned in the late 1980s, is thought to increase the risk of developing PD. Due to its long half-life, it is still present in the soil [248]. In vitro studies indicated that exposure of dopaminergic neurons to dieldrin causes mitochondrial dysfunction, oxidative stress and promotes apoptosis. Kitazawa et. al. showed that, in response to increased ROS production caused by dieldrin exposure, there is an increase in cytochrome C release from mitochondria into the cytosol [249]. Cytochrome C in a complex with apoptotic protease activating factor-1 (Apaf-1) activates caspase-9, one of the prime mover enzymes executing apoptotic cascade [249]. Caspase-9 activates inter alia caspase-3, which is responsible for isoform-specific proteolytic cleavage of protein kinase C gamma type (PKCγ)—a proapoptotic enzyme specific to neurons [249,250].

Another toxin exposure that leads to parkinsonian syndrome is 6-hydroxydopamine (6-OHDA)—a synthetic organic compound used on laboratory animals. 6-OHDA, as a dopamine analogue, is transferred and accumulated in dopaminergic neurons. 6-hydroxydopamine inhibits mitochondrial electron transport chain complex 1. This compound is neurotoxic by itself, unlike MPTP, which has to be oxidized to gain its toxic properties [251]. 6-OHDA inhibits cytochrome C oxidase [251] and, after autooxidation or degradation, generates the ROS responsible for oxidative stress and cell death [252]. 6-OHDA-induced neurotoxicity is widely used in generating animal models of PD. However, the similarity is only partial, as in 6-OHDA neurodegeneration LB are not formed and damage is not present in structures such as the locus ceruleus or in olfactory areas, which is characteristic of PD [253]. 6-hydroxydopamine causes metabolic changes in the striatum; it increases the levels of glutamate and γ-aminobutyric acid (GABA) and decreases the concentration of glutamine, which can be interpreted as a shift in the Gln-Glu cycle between astrocytes and neurons, with possible long-term consequences for equipoise in excitatory and inhibitory brain processes [254].

Manganese (Mn) is a trace mineral vital for the function of many enzymes involved in cellular metabolism. Excess manganese absorbed from the intestine is excreted by the liver, but an overload from inordinate exposure leads to “manganism”, with neurotoxicity caused by Mn accumulation in the basal ganglia. The first report of this phenomenon was made by John Couper, who described the neurologic symptoms among workers at a bleach manufactory subjected to manganese-containing dust [255]. Manganese-induced neurotoxicity has also been described in intravenous drug addicts, who use potassium permanganate as an oxidant in the process of ephedrone synthesis [256].

Mn accumulates mainly in the GP, with an intact dopamine concentration in the SN [257]. Mn influences oxidative phosphorylation causing ROS generation [258]. The highest levels of cellular manganese are found in mitochondria, where Mn^2+^ is attached to the Ca^2+^ binding sites. This leads to the opening of permeability transition pores in the mitochondrial membrane, which causes loss of transmembrane potential, decreases ATP generation and induces apoptosis [259]. Manganese impairs glutamine and GABA uptake, leading to their increased extracellular concentration and neuronal excitability [260,261]; GABA also has an impact on movement regulation in basal ganglia [262], which may have an influence on the clinical picture of manganese-induced neurotoxicity.

The role of oxidative stress in neurodegeneration is beyond doubt; some papers suggest a potential protective effect of urate, a powerful antioxidant, on PD. A meta-analysis conducted by Shen et al. confirmed that a high serum urate level is associated with a 33% reduction in PD development risk; this correlation is dose-related [263]. Another result in that paper pointed to an inverse correlation between the occurrence of gout and PD. Shen et al. found a decline in the PD progression rate associated with increasing serum urate levels. Interestingly, the protective effect of urate applies primarily to the male population, which indicates the potential role of sex hormones [263].

Toxic neurodegeneration leads to Parkinson’s disease-like symptoms. Currently, there is no data indicating an importance of toxins in atypical parkinsonism. The various mechanisms leading to neurodegeneration in PD are shown in Figure 2.

### 5.4. Traumatic Brain Injury as a Neurodegenerative Factor

Concerns about the association between repetitive brain trauma and dementia began more than a century ago but have resurfaced in the last decade as the more recently described chronic traumatic encephalopathy. Chronic traumatic encephalopathy is a tauopathy associated with RBT that has become inextricably linked to conversations about sport-related concussion and mild traumatic brain injury (TBI) [264]. TBI has been stated to be a risk factor for some neurodegenerative disorders such as PD, AD and amyotrophic lateral sclerosis [264,265]. The results from prospective cohort studies indicated an association between TBI and a risk for LB accumulation, parkinsonism and PD [266,267,268]. The most recent study showed that a history of mild TBI is associated with a 56% higher risk of developing PD later in life and the risk grows with increased TBI severity [269].

Amyloid precursor protein (APP), α-synuclein, hyper-phosphorylated Tau and TDP-43 are the most frequently reported proteins upregulated following a traumatic brain injury, which are closely linked to PD [270,271,272,273,274]. Recently, upregulation of LRRK2 has also been linked to TBI in animal models [275,276]. There is a subset of Rab proteins that were identified as biological substrates of LRRK2 linked to late onset PD, whereas inhibition of LRRK2 was found to be a neuroprotective in PD and TBI models [275,276].

## 6. Summary

The clinical symptoms in parkinsonian disorders depend more on the exact brain localization of the damage than the specific, pathophysiological background. There is a lot of evidence that the onset of every neurodegeneration is different, possibly caused by different factors. However, disease progression causes certain processes to overlap and the neuropathological picture includes elements of various pathologies [6]. Increased knowledge in genetics lead to the discovery of two main neurodegenerative processes—protein pathologies and mitochondrial dysfunction. Both of them are highly dependent on inflammation and oxidative stress from the significant impact of environmental factors such as toxins or brain injuries.

At present, however, it is more important to understand the mechanisms differentiating individual processes, because this will be the factor that will allow the use of targeted therapy in each individual. Molecular biomarkers allowing for the differentiation of individual entities belonging to the parkinsonism group will be helpful in this. Genetic factors play an important role here, but the simple diagnosis of α-synuclein is also important, both through easy peripheral biopsies and differentiated α-synuclein in the CSF [142]. The combination of clinical, imaging, genetic and molecular markers in the future will give an early indication as to which neurodegenerative process we are dealing with and allow the implementation of therapies that inhibit the progress of the disease.

## Figures and Tables

**Figure 1 cells-10-00656-f001:**
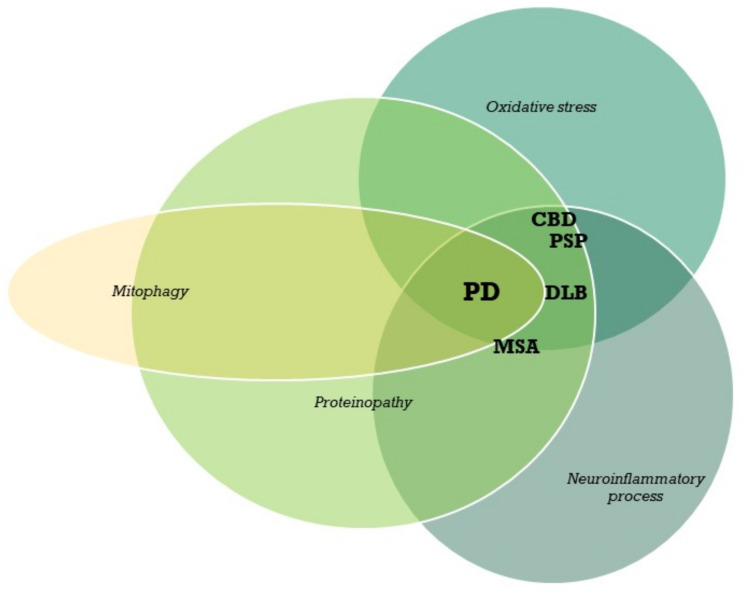
Coexistence of different mechanisms of neurodegeneration in parkinsonism. PD—Parkinson’s disease; CBD—corticobasal degeneration; PSP—progressive supranuclear palsy; MSA—multiple system atrophy; DLB—dementia with Lewy bodies.

**Figure 2 cells-10-00656-f002:**
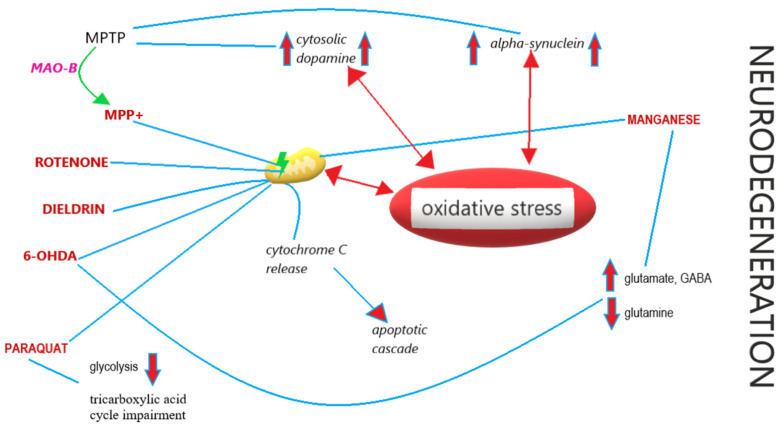
Mechanisms of toxin-induced neurodegeneration.

**Table 2 cells-10-00656-t002:** Genes responsible for the monogenic forms of PD (OMIM).

Locus	Gene	Chromosomal Region	Inheritance	Role of the Encoded Protein
PARK1/4	*SNCA*	4q22.1	AD	-synaptic vesicles mobility
PARK2	*PRKN*	6q26	AR	-mitophagy -ubiquitin-dependent protein degradation
PARK3	---	2p13	AD	-unknown
PARK5	*UCH-L1*	4p13	AD	-participate in generation of ubiquitin monomers
PARK6	*PINK1*	1p36.12	AR	-mitophagy
PARK7	*DJ1*	1p36.23	AR	-redox-sensitive chaperone (sensor for oxidative stress)
PARK8	*LRRK2*	12p12	AD	-chaperone mediated autophagy -participation in neuronal cell death -neuronal plasticity -vesicle trafficking
PARK9	*ATP13A2*	1p36.13	AR	-inorganic cations transport
PARK10	---	1p32	---	-unknown
PARK11	*GIGYF2*	2q37.1	AD	-repressor of translation initiation
PARK12	---	Xq21-q25	X-linked	-unknown
PARK13	*HTRA2*	2p13.1	AD	-caspase-dependent apoptosis
PARK14	*PLA2G6*	22q13.1	AR	-catalyse the release of fatty acids from phospholipids
PARK15	*FBXO7*	22q12.3	AR	-ubiquitination mediator
PARK16	---	1q32	---	-unknown
PARK17	*VPS35*	16q11.2	AD	-retrograde transport of proteins from endosomes to the trans-Golgi network
PARK18	*EIF4G1*	3q27.1	AD	-mRNA binding in translation process
PARK19	*DNAJC6*	1p31.3	AR	-clarithin-mediated endocytosis
PARK20	*SYNJ1*	21q22.11	AR	-regulates levels of membrane phosphatidylinositol-4,5-bisphosphate
PARK21	*DNAJC13/TMEM230*	3q22.1	AD	-co-chaperone of heat shock protein -stimulation of ATP hydrolysis
PARK22	*CHCHD2*	7p11.2	AD	-negative regulator of mitochondria-mediated apoptosis -reduction of oxidative stress
PARK23	*VPS13C*	15q22.2	AR	-proper mitochondrial function (maintenance of mitochondrial transmembrane potential)

**Table 3 cells-10-00656-t003:** Immunological abnormalities in parkinsonisms.

	PD	PSP	MSA	CBD
Aspects of neuro-inflammation in neurodegeneration	(1) Increased concentration of TNF-α in CSF as well as in serum blood (2) Increasedproinflammatory cytokine: IL-6 and IL-12, have been shown to be at increased concentrations in PD serum (3) Increased NT-proCNP (4) PINK1 upregulates IL-1β-mediated signalling (5) Changes in CRP, SAA, IL-8 and YKL-40	(1) Abnormalities in the concentration of cytokines produced by activated microglia as TNF-α, interleukin 1Β and IL-6 (2) Deviations in the regulation of T and Natural Killer Cells related to IL-2	(1) Abnormalities in gene sets of interferon, IL-1, IL-6, IL-8 (2) An increase of CD3^+^, CD4^+^ and CD8^+^ (3) iNOS elevated expression	Via oxidative stress
Neurodegeneration	Oxidative stress, iron metabolism, disturbances, glial mitochondrial dysfunction	Microglial activation affecting tau deposition	Inflammatory response induced by mitochondrial dysfunction	Not explored

## Data Availability

Not applicable.

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
