# Peer review of "Mechanisms of Neurodegeneration in Various Forms of Parkinsonism—Similarities and Differences"

_cells, 2021, doi:10.3390/cells10030656_

Round 1

Reviewer 1 Report

The authors have revised and restructured the manuscript, and have addressed my concerns. The topic is diverse per se, the authors have now managed to present a clearer structure making it easier for the readers to get through and keep the overview. It is a valuable contribution to the field.

Author Response

I wanted to thank you again for the review.

Reviewer 2 Report

The authors have made some of the suggested changes to the manuscript, which has, in some instances, improved the manuscript. However, in most instances the additions have created more confusion. In particular, Table 1 lists the main neuropathological features of different Parkinsonian disorders and describes the type of pathology, macroscopic localisation, microscopic changes, localisation and spreading. Pathology and microscopic changes are to me the same thing, yet for PSP the pathology is described as tau in neurons and the microscopic changes as inclusion bodies in oligodendroglia and astrocytes. The characteristic hallmark of PSP is the tufted astrocyte, which isn’t mentioned anywhere. Similarly, macroscopic localisation and localisation are the same, yet very different anatomical regions are listed under these headings for each disorder. Under DLB, brainstem and SN are listed separately, yet the SN is in the brainstem. The spreading of PSP and CBD is listed as Braak stage but no Braak stage is assigned to PSP or CBD for diagnostic purposes. PNS pathology is described in PD and MSA but confuses regions found in the PNS and CNS. The factual errors and confusing way that this table is presented make it very unhelpful to the reader.

The grammar throughout the manuscript is very poor and the manuscript still lacks logical flow.

Table 3 shows immunological abnormalities in parkinsonisms (should be Parkinsonian disorders) but then a section entitled Neurodegeneration is listed. I am not sure what the purpose of this addition is. The table also does not clearly label what is similar and what is different between the disorders. 

This manuscript does not add sufficiently to the literature and does not further our understanding of mechanisms of degeneration in parkinsonian disorders.

Author Response

Thanks again for your critical comments. The authors wanted to present an overview of the latest literature data, but also their point of view on neurodegenerative processes.

The authors have made some of the suggested changes to the manuscript, which has, in some instances, improved the manuscript. However, in most instances the additions have created more confusion. In particular, Table 1 lists the main neuropathological features of different Parkinsonian disorders and describes the type of pathology, macroscopic localisation, microscopic changes, localisation and spreading. Pathology and microscopic changes are to me the same thing, yet for PSP the pathology is described as tau in neurons and the microscopic changes as inclusion bodies in oligodendroglia and astrocytes. The characteristic hallmark of PSP is the tufted astrocyte, which isn’t mentioned anywhere. Similarly, macroscopic localisation and localisation are the same, yet very different anatomical regions are listed under these headings for each disorder. Under DLB, brainstem and SN are listed separately, yet the SN is in the brainstem. The spreading of PSP and CBD is listed as Braak stage but no Braak stage is assigned to PSP or CBD for diagnostic purposes. PNS pathology is described in PD and MSA but confuses regions found in the PNS and CNS. The factual errors and confusing way that this table is presented make it very unhelpful to the reader.

The inaccuracies in Table 1 have been corrected.

The grammar throughout the manuscript is very poor and the manuscript still lacks logical flow.

The article has been reviewed again by a native speaker.

Table 3 shows immunological abnormalities in parkinsonisms (should be Parkinsonian disorders) but then a section entitled Neurodegeneration is listed. I am not sure what the purpose of this addition is. The table also does not clearly label what is similar and what is different between the disorders. 

Table 3 has been corrected. Similarities and differences are highlighted.

This manuscript does not add sufficiently to the literature and does not further our understanding of mechanisms of degeneration in parkinsonian disorders.

The continuous clinical division of disease entities does not primarily result from neuropathological differences, but come mainly from the different locations of the process. On the other hand, we know that in the same disease we have different neurodegenerative processes and different genetic factors. This work provides a systematic review of various concepts for neurodegenerative processes at the molecular (genetics), cellular (autophagy / mitophagy) and tissue (inflammatory processes / oxidative stress) levels. We also refer to the role of toxic factors or brain injuries in the above-mentioned neurodegenerative processes.

This manuscript is a resubmission of an earlier submission. The following is a list of the peer review reports and author responses from that submission.

Round 1

Reviewer 1 Report

In general the manuscript is difficult to read, lacks flow and is insufficiently referenced. The introduction of each of the disorders is inconsistent and it is unclear what the purpose of this introduction is. For example, each disorder should be introduced according to its prevalence, average age of onset, clinical symptoms, neuropathology. Whilst this is provided for some of the diseases (PD), others are less comprehensively introduced (ie DLB). Furthermore, the description of the pathology is incomplete, with no reference to the neuropathological diagnostic criteria. There is also no clear distinction between a clinical diagnosis and the neuropathological diagnosis for the most part, which we know can be very different, especially for disorders such as CBD.

The title of this review does not reflect its content. A comprehensive look at the similarities and differences between these disorders and why they produce similar clinical symptoms would require a critical appraisal of the pathology and mechanisms of degeneration in relation to their anatomical distribution and the clinical features of the disease. The article does not discuss the possibility that different pathologies may give rise to similar clinical phenotypes by virtue of the fact that they cause degeneration in specific brain regions associated with that function and/or how changes in neuroinflammation, mitophagy etc are specifically altered in these regions. The authors instead focus on tau and alpha-synuclein but this is not specifically relevant to parkinsonism itself. Rather, tau pathology is also found in Pick’s disease, globular glial tauopathy, Alzheimer’s disease etc but these disorders are not reviewed. I am therefore left confused about the articles main focus.

The author states that DLB can masquerade as MSA but DLB also commonly masquerades as AD but there is little mention of this. This selective presentation of the literature is very misleading and unhelpful to the reviewer.  

The grammatical flow of the article requires attention. In particular, new paragraphs are started where they are not required (many paragraphs are just a few lines long). This makes the review very hard to read. The manuscript also reads like a list of facts rather than a comprehensive review of the current literature in places.

The summary does not reflect the aim of the review. In particular, it focusses on diagnosis and alpha-synuclein but there is no mention of oxidative stress or neuroinflammation etc.

On page 15 the authors state –

A glycoprotein involved in inflammation, progranulin, which is associated with the GRN gene, was interpreted as possible being correlated with frontotemporal lobar degeneration, a possible pathology related to corticobasal syndrome.

This leaves me confused as we now appear to be confusing the clinical syndrome with the neuropathological disease and the authors have introduced FTLD-TDP-43.

In Table 2 the authors also interchange CBD with CBS. CBD is a tauopathy that can present as CBS or may also present as other clinical phenotypes,  such as Alzheimer’s. CBS is a clinical disorder only.

The authors state that 3 and 4 R tau isoforms characterise Pick’s disease but Pick’s disease is primarily a 3R tauopathy.

The section on TBI is confusing and is not well integrated into this review. There is also no mention of it in the abstract or summary. There is also no mention of CBD in the abstract. This leaves the reader confused about the focus of the article.

Reviewer 2 Report

The review of Koziorowski et al. aims to address differences and similarities of various forms of parkinsonism. 

Due to the nature of the topic, it is a very complex manuscript, trying to cover a lot of aspects. The review starts to introduce the different forms of parkinsonisms, and then provides information about genetics, protein pathologies, followed by a paragraph entitled mitophagy and a comprehensive last paragraph about neurodegenerative factors. The latter is subdivided in oxidative stress, neuroinflammation, toxic neurodegeneration and traumatic brain injury.

Together, this review is very comprehensive, but also very confusing and needs to be more clearly structured and/or focused. Bundling of information, maybe focussing on less aspects, more informative headings and more schemes/tables might help.

The part “Genetics of parkinsonism“ could be better structured. Including the known function of genes depicted in Table 1 in this table would provide a better overview. Also the occurrence in percent in PD patients should be included. Subheadings for the different diseases might further help.

The  concept of the third paragraph: „Protein pathology in various forms of parkinsonism.“ needs to be structured more clearly by precise headings or even more subheadings.

In paragraph 3.2. (Mechanisms of α-synuclein toxicity), there is a small paragraph dealing with α -synuclein in the repair of nuclear DNA. First, this paragraph seems incomplete. The underlying mechanisms could be described more in detail, if known. As it stands it appears displaced.

A scheme about all these mechanisms affected by α -synuclein could provide the overview. Further, as α -synuclein is also localised in synapses, the association of synaptic function regulation and survival could be highlighted more.

The headline of the 4th paragraph: „Mitophagy“ should be more precise. What do the authors want to tell us here? This should be already clear from reading the subheading. For me, this is not clear!

Likewise imprecise is the heading of the 5th paragraph: „Various neurodegenerative factors“. What are neurodegenerative factors? Under the first subheading „5.1. oxidative stress“, a large bunch of information is collected, again relatively confusing, of which the iron-associated studies are comprehensively described. Do they might deserve an own paragraph? Here the authors jump between the different forms of parkinsonism, which makes it hard to follow and to extract important information. The reader is flashed with details that must be better structured.

Here, already the connection between oxidative stress, iron metabolism and neuroinflammation is mentioned, whereby neuroinflammation is more detailedly described in the next paragraph: 5.2. Neuroinflammation as a factor in neurodegeneration of parkinsonisms.

Here, again, what means factor? What stays unclear to me is, what do the authors judge as cause or consequence? This becomes especially important for neuroinflammation. Neuroinflammative processes are seen in many neurodegenerative diseases. Are they more causal or secondary effects that drive certain aspects of neurodegeneration?

As the review claims to compare the different forms of parkinsonism, would it maybe better instead of subdividing in „oxidative stress“ and „neuroinflammation“, processes, which cannot be seen independently, to separate mir in terms of  what is known for the different forms of parkinsonism? I think, this is an important paragraph, which however needs to be better structured.

The following phrase: „The role of oxidative stress in neurodegeneration is beyond doubt; some papers suggest a potential protective effect of urate, a powerful antioxidant, on PD. Meta-analysis conducted by Shen et al. confirmed that a high serum urate level is associated with a33% reduction in PD development risk; this correlation is dose-related [276]….“ is collected in „5.3. Toxic neurodegeneration.“, wouldn’t it better be discussed, when discussing oxidative stress? seems displaced now. in this paragraph, the authors mention another toxin exposure which leads to parkinsonian syndrome: 6-hydroxydopamine (6-OHDA). Where does it come from? where can people be exposed to this toxin?

Could the crosstalk of inflammation, iron und toxicity be graphically captured?

In the summary, the autors claim: „There is a lot of evidence that the onset of every neurodegeneration is different, possibly caused by different factors. However, disease progression causes certain processes to overlap and the neuropathological picture includes elements of various pathologies [6].“

This is exactly, what need to be made more clear: what are the causes? How do they differ? what are the „secondary“ mechanisms and how do they differ/overlap. As the review is written now, no clear picture emerges.

better structure of 5.1. Oxidative stress —> better headlines better collecting the different issues, e.g., Contribution of oxidative stress-mediated disturbed iron homeostasis; the whole paragraph must be better structured, confusing. A scheme, collecting the most relevant information would help!

minor points:

-control space before reference numbers

-sometimes punctuation is missing

-acheck spelling